# Highly Thermal Conductive and Electrically Insulating Epoxy Composites Based on Zinc-Oxide-Coated Silver Nanowires

**DOI:** 10.3390/polym14173539

**Published:** 2022-08-29

**Authors:** Li Zhang, Wenfeng Zhu, Guoquan Qi, Houbu Li, Dongtao Qi, Shuhua Qi

**Affiliations:** 1State Key Laboratory of Performance and Structural Safety for Petroleum Tubular Goods and Equipment Materials, CNPC Tubular Goods Research Institute, Xi’an 710077, China; 2Department of Applied Chemistry, School of Chemistry and Chemical Engineering, Northwestern Polytechnical University, Xi’an 710072, China

**Keywords:** silver nanowires, zinc oxide, thermal conductivity, epoxy composite

## Abstract

Nano-ZnO particles were deposited on the surface of silver nanowires (AgNWs) by the precipitation method, and the obtained AgNW@ZnO filler with core-shell structure was added to epoxy resin to prepare AgNW@ZnO/EP thermally conductive composites. The ZnO coating on the surface of AgNWs effectively improves the thermal conductivity of the composites. When 8 wt% AgNW@ZnO filler was added to the epoxy resin, the thermal conductivity of the AgNW@ZnO/EP composite increased to 0.77 W/(m·K). The enhancement of the thermal conductivity is attributed to the fact that ZnO effectively improves the interface bonding between AgNWs and the epoxy matrix, thus strengthening the contact between AgNWs. In addition, the electrical insulation of the AgNW@ZnO/EP composites was improved upon the introduction of the ZnO coating over AgNWs. For the filler content of 8 wt%, the volume resistivity of the AgNW@ZnO/EP composites was found to be higher than 10^13^ Ω·cm. AgNW@ZnO/EP composites have also exhibited low dielectric constant and good thermal stability.

## 1. Introduction

The rapid developments in the electronics industry show that the miniaturization of electronic equipment is the future trend. As the miniaturized electronic equipment is likely to produce an enormous amount of heat during operation, heat dissipation therein has become a critical issue. If the generated heat is not dissipated in time, the internal temperature will rise sharply, which affects the reliability, stability, and lifespan of the electronic components. Generally, in integrated circuits, heat sinks are used to get rid of the heat produced by the chip. However, the gaps between the chips and the heat sinks may hinder the efficient transfer of heat, thus resulting in extremely low heat dissipation efficiency by the heat sinks. To solve this problem, thermal interface materials (TIMs) with high thermal conductivity need to be used to fill the gaps to reduce the interfacial thermal resistance. Polymers are widely used as TIMs due to their low cost, light weight, chemical resistance, and ease of processing. Ceramic fillers such as Al_2_O_3_ [1,2,3], BN [4,5,6], AlN [7,8,9], and SiC [10,11,12] have been added to polymers to improve the thermal conductivity of polymer composites. However, large amounts of ceramic filler content (30~60 vol%) are needed to achieve higher thermal conductivity, which eventually shows a negative impact on the mechanical properties and processability of the composites.

One solution to the above problem is to use nanofillers with high thermal conductivity. Compared with ceramic fillers, metallic silver (Ag) has extremely high thermal conductivity. Its thermal conductivity at room temperature is 427 W/(m·K), which is much higher than the ceramic fillers used currently. Silver nanowires (AgNWs) are one-dimensional structures that can be used as metallic fillers. AgNWs not only inherited the excellent thermal conductivity of their bulk form but also have a large specific surface area, showing great application potential in the field of thermally conductive materials. AgNWs with a high aspect ratio have the ability to construct a long heat conduction path in the lateral direction. Therefore, the thermal conductivity percolation threshold of the composite material with AgNWs fillers can be significantly reduced, and thermal conductivity can be effectively improved even at the lower filler content [13].

In addition, AgNWs are also excellent electrical conductors. The electrical resistivity of silver is 1.59 × 10^−6^ Ω·cm, which is the lowest among all the metals. Therefore, the addition of AgNWs into the polymer matrix may result in a decrease in the electrical insulation properties. TIMs must have high thermal conductivity and high electrical insulation. Thus, the high electrical conductivity of AgNWs limits their applications to develop TIMs. However, surface modification of AgNWs can be carried out to improve the electrical insulation characteristics of the composites filled with AgNWs.

According to previous literature reports, the surface modification of metal nanowires is carried out using common inorganic materials, SiO_2_ and TiO_2_. Simple preparation methods and mild reaction conditions of SiO_2_ and TiO_2_ make them suitable for the surface modification of other nanomaterials, such as CNTs [14] and graphene [15]. However, the thermal conductivity of SiO_2_ and TiO_2_ is very low, about 1.5~3 W/(m·K) [16] and 2.5~5 W/(m·K) [17], respectively. Depositing inorganic oxides with low thermal conductivity on the surface of AgNWs cannot effectively take advantage of the high thermal conductivity of AgNWs.

On the other hand, nano-ZnO (particle size of 5–100 nm) is a versatile multifunctional inorganic material. Due to its unique properties, it is explored in many fields, such as catalysis, optics, magnetism, and mechanics. Further, ZnO is also used as a thermally conductive filler, owing to its good thermal conductivity of 60 W/(m·K) (which is much higher than that of SiO_2_ and TiO_2_). Moreover, nano-ZnO shows good electrical insulation characteristics, with its resistivity in the range of 10^6^~10^9^ Ω·cm [18]. Therefore, surface modification of AgNWs with nano-ZnO can facilitate exploitation of the high thermal conductivity of AgNWs. At the same time, the formation of an insulating layer over the surface can effectively reduce the electrical conductivity of AgNWs, thereby enhancing the electrical insulation of the composites.

In this paper, AgNWs with a high aspect were prepared. The morphology and microstructure of the AgNWs were characterized. Subsequently, the precipitation method was used to deposit nano-ZnO over the surface of AgNWs, thus obtaining ZnO-coated AgNWs (AgNW@ZnO) filler. AgNW@ZnO was added to epoxy (EP) resin to prepare AgNW@ZnO/EP composite material. Thermal conductivity, resistivity, thermal stability, and dielectric constant of the AgNW@ZnO/EP material were studied and results were compared with the unmodified AgNW/EP composites. The observed differences and associated mechanisms were discussed and analyzed.

## 2. Experimental

### 2.1. Materials

Ethanol used as solvent was obtained from Tianjin Fuyu Chemical Co. (Tianjin, China). Sodium chloride (NaCl) and glycerol, used as nucleating and reducing agents for AgNWs synthesis, were also obtained from Tianjin Fuyu Chemical Co. (Tianjin, China). Polyvinylpyrrolidone (PVP, *M*_w_ = 400,000), a surfactant used in the synthesis of AgNWs, was obtained from Tokyo Chemical Industry Co. (Tokyo, Japan). Silver nitrate (AgNO_3_), used to synthesize AgNWs, was obtained from Aladdin Bio-Chem Technology Co. (Shanghai, China). Sodium hydroxide (NaOH) and zinc nitrate hexahydrate (Zn(NO_3_)_2_·6H_2_O) used for the synthesis of nano zinc oxide were obtained from Guangdong Guanghua Chemical Co., Ltd. (Shantou, China). Epoxy resin (E51) was obtained from Zhejiang Materials Industry Chemical Group Co. (Hangzhou, China), and the corresponding molecular structure is shown in Figure 1. The molecular weight and epoxy equivalent weight of epoxy resins are 370~420 and 190, respectively. The curing agent m-xylylenediamine (MXDA) was supplied by Macklin Chemistry Co. (Shanghai, China).

### 2.2. Preparation of Silver Nanowires (AgNWs)

AgNWs were prepared by reducing AgNO_3_ with glycerol at high temperature as follows. First, 5.85 g PVP was added to 190 mL glycerol in a round bottom flask at 50 °C, and mechanical agitation was used to ensure homogeneity of the solution. When the solution was cooled to room temperature, 1.58 g AgNO3 was added to it. A mixed solution of 10 mL glycerol and 0.5 mL deionized water containing 59 mg of NaCl was then added to the flask. The solution was continuously stirred for 1 h at 210 °C, and then it was cooled naturally to room temperature. Finally, the resulting solution was diluted by adding 200 mL of H_2_O, followed by centrifugation at 8000 rpm for 30 min to obtain AgNWs. The centrifugation process was repeated three times by using ethanol to remove residual glycerol and PVP on the surface of AgNWs. The obtained AgNWs were then dispersed in ethanol solution for later use at a concentration of 0.0053 g/mL.

### 2.3. Preparation of AgNW@ZnO

There exist many methods to synthesize nano-ZnO in the literature, which include solid-phase reaction [19], hydrothermal process [20], precipitation technique [21], chemical vapor deposition [22], electrolysis [23], and magnetron sputtering [24]. Among these methods, the precipitation technique is most widely used due to its simple operation and mild reaction conditions. The basic principle is to add the precipitant dropwise to the salt solution containing zinc to slowly precipitate crystal nuclei at an appropriate temperature. Further growth of nuclei results in nano-scale ZnO crystal grains. Therefore, in this paper, the precipitation method was used to synthesize nano-ZnO over the surface of the prepared AgNWs. The schematic diagram of the preparation process of AgNW@ZnO is shown in Figure 2.

The typical preparation process of AgNW@ZnO is as follows: First, 100 mL ethanol solution containing AgNWs with a concentration of 0.0053 g/mL was added to the reactor. Then 0.44 g Zn(NO_3_)_2_·6H_2_O was added and stirred for 1.5 h to fully dissolve. In addition, 0.11 g NaOH was dissolved in 100 mL ethanol by stirring for 2 h. The prepared ethanol solution containing NaOH was added dropwise to the solution containing AgNWs. The process was carried out slowly such that the ethanol solution with NaOH was added dropwise within 1 h. High-speed stirring was maintained throughout the reaction process for the next 2 h. Next, the reaction vessel was sealed and equilibrated for 24 h. Upon complete precipitation of the product, the supernatant was removed and the remaining solution was centrifuged at 4000 rpm for 10 min. The obtained precipitate was separated and washed several times with deionized water and ethanol to remove the residues in the reaction. The final product was dried in a vacuum oven at 60 °C for 12 h to obtain AgNW@ZnO filler.

### 2.4. Preparation of Composite Materials

The composite material was prepared by the solution mixing method. AgNW@ZnO filler to a certain proportion was completely dispersed in ethanol solution ultrasonically for 20 min followed by stirring for 1 h. Subsequently, the solution was heated to 70 °C, and a certain amount of epoxy resin was slowly added to the solution, which was ultrasonicated for 20 min to ensure thorough mixing. The obtained mixture was vacuum-dried at 70 °C to remove the residual ethanol from it. Further, the curing agent m-xylylenediamine (MXDA) was added to the epoxy mixture, which was 25 wt% of the epoxy resin, and stirred for 10 min. The final epoxy resin product filled with AgNW@ZnO filler was vacuum-dried and cured at 120 °C for 8 h to obtain the AgNW@ZnO/EP composite.

### 2.5. Material Characterization

X-ray diffractometer (XRD; XRD-6000, Shimadzu, Kyoto, Japan) was used to characterize the crystal structure of the synthesized AgNWs and AgNW@ZnO. The morphologies of AgNWs and AgNW@ZnO and their distribution in the epoxy matrix were examined using scanning electron microscopy (SEM; Verios G4, FEI, Hillsboro, OR, USA). The distribution morphology of nano-ZnO on the surface of AgNWs was observed by transmission electron microscopy (TEM; Talos F200X, FEI, Hillsboro, OR, USA). The static contact angle between the filler and the epoxy matrix was measured using a static contact angle meter (OCA20, Stuttgart, Germany). Thermogravimetric analysis (TGA) of epoxy, AgNW/EP, and AgNW@ZnO/EP composites was performed using a thermal analyzer (Q5000IR, TA Instruments, New Castle, DE, USA). During the test, the temperature was increased from 25 °C to 700 °C in an air atmosphere at a heating rate of 10 °C/min. The glass transition temperatures (*T*_g_) of epoxy resin, AgNW/EP, and AgNW@ZnO/EP composites were measured using a differential scanning calorimeter (DSC; Q2000, TA Instruments, New Castle, DE, USA). The volume resistivity of AgNW/EP and AgNW@ZnO/EP composites was measured using an ultra-high resistivity meter (ZC36, Shanghai, China). The dielectric constant of composites was tested by dielectric constant tester (ZJD-C, Beijing, China). The test frequency range was 10^5^ Hz~10^7^ Hz. The diameter of the tested sample was 38.4~40 mm and the thickness was controlled between 1 and 5 mm. The thermal conductivity of epoxy, AgNW/EP and AgNW@ZnO/EP composites was determined by a laser flash analyzer (LFA 447, Netzsch, Bavaria, Germany). The surface temperature of the composites was measured using an infrared camera (IRS-S6, Shanghai, China).

## 3. Results and Discussion

### 3.1. Characterization of AgNWs

AgNWs were synthesized using the polyol method by heating at 210 °C for 1 h [25] as discussed above. Compared with the synthetic method which used ethylene glycol, glycerol contains more hydroxyl groups and therefore acts as a stronger reducing agent. Figure 3 shows SEM images and XRD pattern of the obtained AgNWs. It can be seen from SEM images that the prepared AgNWs have a uniform diameter and contain fewer particulate impurities. The purified AgNWs are one-dimensional nanostructures with a length of 10–50 μm and a ratio of a diameter ranging from 100–130 nm. A large number of AgNWs are piled up to form a random network structure, which is conducive to the formation of phonon transmission channels in the polymer matrix, thereby effectively reducing the thermal resistance. Five diffraction peaks at 38.1°, 44.3°, 64.4°, 77.4°, and 81.5° are noted in the XRD patterns of AgNWs, which correspond to the (111), (200), (220), (311), and (222) crystal planes of face center cube crystals [26]. Thus, the successful preparation of AgNWs is confirmed.

### 3.2. Characterization of AgNW@ZnO

Nano-ZnO particles can be obtained by controlling the ratio of zinc salt and the precipitation agent in the precipitation method [27]. In this paper, by adjusting the ratio and concentration of Zn(NO_3_)_2_·6H_2_O and NaOH, nano-ZnO was coated over the AgNWs, which is termed AgNW@ZnO. SEM images, TEM images, and XRD pattern of the AgNW@ZnO are shown in Figure 4a–d, respectively.

SEM image in Figure 4a clearly shows AgNWs covered by tiny ZnO particles. The coating of ZnO is relatively complete, and there are no exposed AgNWs. The thickness of ZnO coating is uniform, and no large agglomerates of ZnO are found. Figure 4b is the high-magnification SEM image of AgNW@ZnO filler, in which nano-ZnO particles formed by stacking over the surface of AgNWs can be observed clearly. The diameter of nano-ZnO particles is 5–25 nm, and the thickness of ZnO coating is 15–50 nm as seen in the TEM image of AgNW@ZnO (Figure 4c). By comparing with the SEM images of pure AgNWs, it can be observed that the surface of the AgNW@ZnO with core-shell structure is much rougher. Rough surface AgNW@ZnO fillers have a larger specific surface area, which is beneficial to obtain good dispersion in the matrix.

The crystalline nature of AgNW@ZnO was characterized by an XRD study, and the results are shown in Figure 4d. The diffraction peaks of (111), (200), (220), (311), and (222) crystal planes corresponding to the AgNWs are seen. In addition, typical characteristic diffraction peaks of ZnO appeared at 31.9°, 34.5°, 36.3°, 47.7°, 56.7°, 62.9°, and 68.1° in the XRD pattern, which can be indexed respectively to the (100), (002), (101), (102), (110), (103), and (112) crystal planes of the hexagonal wurtzite structure of ZnO. Except for AgNWs and ZnO phase diffraction peaks, no other peaks are noticed, which indicates the high purity of the obtained AgNW@ZnO filler.

### 3.3. Static Contact Angle Test of AgNW@ZnO

The prepared AgNWs and AgNW@ZnO fillers were respectively coated on glass plates to form thin films, and their static contact angles with epoxy resin were estimated. Images of the contact angle test are shown in Figure 5, and the contact angle data are given in Table 1.

The contact angle of AgNW filler with epoxy resin is 31.7°, whereas, for AgNW@ZnO, it is 13.2°. Upon coating AgNWs with the nano-ZnO layer, the contact angle of the filler to epoxy decreased by 18.5°, which indicates the enhancement of the interfacial bonding of filler with epoxy matrix. The observed differences may be due to the smooth surface of pure AgNWs, which leads to the relatively poor wettability by epoxy. On the other hand, the surface of AgNWs becomes rough upon nano-ZnO coating, which is beneficial to enhance the wettability of epoxy resin to fillers. In addition, nano-ZnO prepared in solution possesses a large number of hydroxyl groups on its surface [28,29,30,31]. These hydroxyl groups can interact with epoxy resin in a certain manner, which helps to enhance the wettability of epoxy resin to the filler. Improving the wettability of resin and filler enhances the interfacial bonding between them.

### 3.4. Characterization of Composite Materials

For nano-sized AgNW@ZnO filler, it is very important to ensure its uniform dispersion in the matrix [32,33]. Since the prepared AgNW@ZnO filler can disperse better in ethanol, ethanol was used as a solvent to mix the prepared nanofillers into epoxy resin to prepare composite materials. In order to investigate the distribution of fillers in composites, SEM was used to characterize the cross-sections of epoxy composites filled with AgNWs and AgNW@ZnO. The cross-section SEM images of the composites are shown in Figure 6. Figure 6a,b are the cross-sectional SEM images of AgNWs-filled epoxy resin composites, whereas Figure 6c,d are the SEM images of AgNW@ZnO-filled epoxy composites cross-sections.

It can be seen that, in Figure 6a,c, there is no obvious difference in the filler distribution of two epoxy composites, with AgNWs and AgNW@ZnO. Both fillers are uniformly dispersed in epoxy resin, and no agglomeration is noticed. Therefore, the use of ethanol as a solvent ensures good dispersion of the fillers in epoxy resin. However, in Figure 6b, gaps at the interface of some AgNWs and epoxy matrix indicate that AgNWs are poorly bound to the epoxy matrix. This may be because of the poor interface wettability of epoxy to AgNWs, resulting in some AgNWs not closely interacting with the epoxy matrix. Whereas, no such cases are observed in AgNW@ZnO-filled epoxy resin. Figure 6d shows a high-magnification SEM image of the AgNW@ZnO distribution in an epoxy matrix. AgNW@ZnO is embedded into the epoxy matrix, and there are no gaps at their binding interface. The close binding of AgNW@ZnO and epoxy matrix helps to reduce the interfacial thermal resistance caused by poor contact between them and effectively improves the thermal conductivity of the composite material.

### 3.5. Thermal Conductivity of Composites

The thermal diffusivity of the AgNW/EP and AgNW@ZnO/EP composites was measured using the laser thermal conductivity meter, and the corresponding thermal conductivities of the composites materials were calculated. The results are shown in Figure 7.

The thermal conductivity of the AgNW/EP composites is significantly improved and increased with the increase of AgNWs content. For instance, the thermal conductivity of pure epoxy resin is only 0.18 W/(m·K), whereas, for 8 wt% AgNWs fillers content, AgNW/EP composites’ thermal conductivity is increased to 0.63 W/(m·K). A similar increasing trend of thermal conductivity is observed for AgNW@ZnO filler. In contrast, for the same mass fractions of two fillers, the thermal conductivity of AgNW@ZnO/EP composite is found to be higher than that of AgNW/EP composite. At 8 wt% AgNW@ZnO filler content, the thermal conductivity of AgNW@ZnO/EP composites is 0.77 W/(m·K), which is 22% higher than the AgNW/EP composites.

From the thermal conductivity curves of AgNW/EP and AgNW@ZnO/EP composites, it can be seen that the overall thermal conductivity of the composites is increased for the composites with nano-ZnO-modified AgNWs fillers, though ZnO has low thermal conductivity. This may be explained by the following two factors: as explained above, the first one is related to the interfacial binding of filler-epoxy. As the surface of pure AgNWs is very smooth, the wettability of epoxy resin to the surface of AgNWs is relatively poor. This results in a small fraction of air gaps or voids at their interface and creates huge interfacial thermal resistance, due to the very low thermal conductivity of air, which is only 0.024 W/(m·K). It affects the overall thermal conductivity of AgNW/EP materials. On the other hand, the surface of AgNW@ZnO is relatively rough and it accompanies a large number of hydroxyl groups on its surface, which have a certain interaction with the epoxy matrix. AgNW@ZnO is bonded intimately with the epoxy matrix, which reduces the interfacial thermal resistance and enhances the overall thermal conductivity of the composite. The thermal conduction of AgNWs in the composite mainly depends on the thermal movement of free electrons, and its heat transfer efficiency is very high. The heat conduction of the resin matrix mainly depends on the thermal vibration of the molecular chains, and its heat transfer efficiency is extremely low. Good interface bonding between the filler and the resin is beneficial to reduce the scattering of electrons and phonons at the interface for better heat transfer efficiency.

In addition, during the preparation of AgNW@ZnO fillers, ZnO is not only deposited over the surface of AgNWs but is also deposited over the contact points between AgNWs. Due to this, the loose network originally formed by the overlapping of AgNWs is strengthened by the deposited ZnO at the contact points. The strengthened AgNWs network further facilitates the stable transport of phonons and electrons in it, resulting in higher thermal conductivity in the composites.

### 3.6. Electrical Properties of Composites

Volume resistivity of the AgNW/EP and AgNW@ZnO/EP composites was measured, and the results are shown in Figure 8. The volume resistivity of the AgNW/EP is decreased sharply with the increase of AgNWs filler content. For 8 wt% AgNWs, the volume resistivity dropped by seven orders of magnitude. However, the volume resistivity of AgNW@ZnO/EP composites decreased gradually with the increase of filler content. For 8 wt% AgNW@ZnO filler fraction, the volume resistivity of the AgNW@ZnO/EP composites remains as high as 10^13^ Ω·cm, which indicates good electrical insulation. As AgNWs are one-dimensional nanofillers with excellent electrical conductivity, increasing the mass fraction of AgNWs forms a local conductive network which significantly reduces the overall volume resistivity of the AgNW/EP composites. Whereas, the resistivity of nano-ZnO lies between 10^6^ and 10^9^ Ω·cm, which is much larger than the AgNWs. The nano-ZnO coating over the surface of AgNWs hinders the direct contact between AgNWs, which restricts the electron transport of free electrons within the single AgNW. The nano-ZnO coating thus breaks the conductive network path formed by the pure multiple AgNWs at higher fractions. Consequently, the electrical insulating properties of the AgNW@ZnO/EP composites are significantly enhanced. The AgNW@ZnO/EP thermally conductive composite material can be recommended for specific applications where high electrical insulation is also required.

### 3.7. Thermal Management Capabilities of Composites

In order to investigate the heat dissipation performance of different composites, circular samples of pure EP, AgNW/EP, and AgNW@ZnO/EP composites were placed on a heating platform with an initial temperature of 30 °C. Subsequently, the platform heats up with constant power to simulate the heat dissipation of electronic equipment. The infrared thermal imager is used to monitor the surface temperature of the composite samples. Thermal images were taken at every 20 s interval as given in Figure 9. The filler content in AgNW/EP and AgNW@ZnO/EP composites used for this study is 8 wt%.

The surface temperature of all the samples is increased with the increase of heating duration as shown in temperature curves (Figure 9); however, the rate of temperature rise is different in each of them. Among them, pure EP has shown the slowest temperature rise because of its low thermal conductivity, which results in extremely slow heat transfer. As the thermal conductivity of the composites is increased upon the addition of thermally conductive fillers, the temperature rise rate is significantly improved. Further, the temperature increase rate of AgNW@ZnO/EP composite is found to be higher than that of AgNW/EP composite, which is due to the better thermal conductivity of AgNW@ZnO/EP composites. The infrared thermal photographs that were taken during different time intervals also intuitively reflect the better thermal conductivity of the AgNW@ZnO/EP composite.

### 3.8. Thermal Stability of Composites

Since thermally conductive materials are often used in high-temperature applications, they should possess good thermal stability. Therefore, the thermal stability of thermally conductive materials is an important parameter to evaluate their comprehensive performance.

Thermogravimetric analysis (TGA) and differential scanning calorimetry (DSC) were used to characterize the thermal stability of EP, AgNW/EP, and AgNW@ZnO/EP samples. The corresponding results are shown in Figure 10. The filler content of AgNW/EP and AgNW@ZnO/EP composites with 4 wt% is used for this test. T_5%_ and T_50%_ are used as the parameters to characterize the thermal weight loss of the samples, which represent the temperature corresponding to the material mass loss of 5% and 50%, respectively. T_5%_ is also referred to as the initial decomposition temperature, which can be used to characterize the heat resistance of materials. T_5%_, T_50%,_ and glass transition temperature (*T_g_*) of pure EP, AgNW/EP, and AgNW@ZnO/EP samples are listed in Table 2.

Initial weight of the samples is decreased gradually during the heating process, and the weight is dropped rapidly upon reaching a certain temperature. T_5%_ and T_50%_ of pure EP are 271.8 °C and 372.1 °C, respectively, whereas, for AgNW/EP, T_5%_ and T_50%_ are increased to 300.2 °C and 374.9 °C, respectively. Further, for AgNW@ZnO/EP, T_5%_ and T_50%_ are 329.4 °C and 382.3 °C, respectively which are significantly higher as compared with pure EP. The significant improvement in thermal stability of AgNW@ZnO/EP composites may be due to the fact that the ZnO coating over the surface of AgNW increases the interfacial bonding between the fillers and matrix, thereby increasing the activation energy of the thermal decomposition process in the composites.

The thermal stability of pure EP, AgNW/EP, and AgNW@ZnO/EP were further characterized by DSC. *T_g_* of the EP is enhanced upon adding AgNWs and AgNW@ZnO to it, by about 5 °C and 8 °C, respectively, as shown in DSC curves (Figure 10). The DSC data also indicate the higher thermal stability of the AgNW@ZnO/EP composite.

### 3.9. Dielectric Constant of Composites

The dielectric constants of pure EP, AgNW/EP, and AgNW@ZnO/EP composites with 2 wt% filler content composites were determined in the high-frequency range (10^5^ Hz–10^7^ Hz). The result is shown in Figure 11.

Dielectric constant of pure EP is low, and it is 2.98 at the electric field frequency of 10^5^ Hz (as shown in Figure 11). Whereas, the dielectric constant of AgNWs/EP and AgNW@ZnO/EP composites is significantly increased. AgNWs are metallic materials with very good electrical conductivity, which is very different from that of epoxy resin. Under the external electric field, a large amount of charge accumulation takes place at the metal-epoxy interfaces, resulting in large interfacial polarization and dielectric constant. Further, the dielectric constant of AgNW@ZnO/EP is found to be larger than that of AgNW/EP. This is again attributed to the better filler and matrix interface in the case of nano-ZnO-coated AgNWs. This enhances the polarization at the interface junction, which leads to the increase in the dielectric constant of the AgNW@ZnO/EP composite.

## 4. Conclusions

In this paper, AgNWs with one-dimensional structure were prepared and nano-ZnO particles were grown over the surface of AgNWs using the precipitation method to obtain AgNW@ZnO fillers with core-shell structure. Thermally conductive AgNW@ZnO/EP composites were prepared by adding AgNW@ZnO into epoxy resin. Compared with AgNW/EP composite filled with the same quantity of AgNWs, the thermal conductivity of AgNW@ZnO/EP composite is higher. AgNW@ZnO filler has shown a better effect on improving the thermal conductivity of epoxy resin than pure AgNWs filler. For 8 wt% filler content of AgNW@ZnO, the thermal conductivity of the composite material is increased to 0.77 W/(m·K). The surface of AgNW@ZnO is rougher than the AgNWs and contains a large number of hydroxyl groups over the surface. This contributes to the tight bonding between AgNW@ZnO and epoxy matrix, which reduces the filler-epoxy interface voids. It further leads to the lowering of interface thermal resistance, thereby improving the thermal conductivity of the epoxy composite.

An electrically insulating nano-ZnO coating over the surface of AgNWs also helped in preventing the free electrons from efficiently transporting between the networks formed by AgNWs. This inhibits the formation of the local conductive network in the composite, thus enhancing the insulation of epoxy composite. For instance, for 8 wt% AgNW@ZnO filler content, the volume resistivity of AgNW@ZnO/EP composite is still higher than 10^13^ Ωcm, indicating that AgNW@ZnO/EP composites possess good electrical insulation. Further, AgNW@ZnO-filled epoxy composites have shown higher initial thermal decomposition temperature, glass transition temperature, and thermal stability than AgNWs-filled epoxy composites. In addition, ZnO coating over the AgNWs does not significantly change the dielectric constant of epoxy composites, and still shows a low dielectric constant.

## Figures and Tables

**Figure 1 polymers-14-03539-f001:**
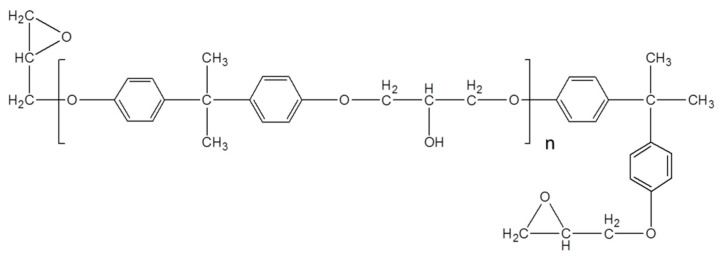
The molecular structure of the epoxy resin used in this paper.

**Figure 2 polymers-14-03539-f002:**
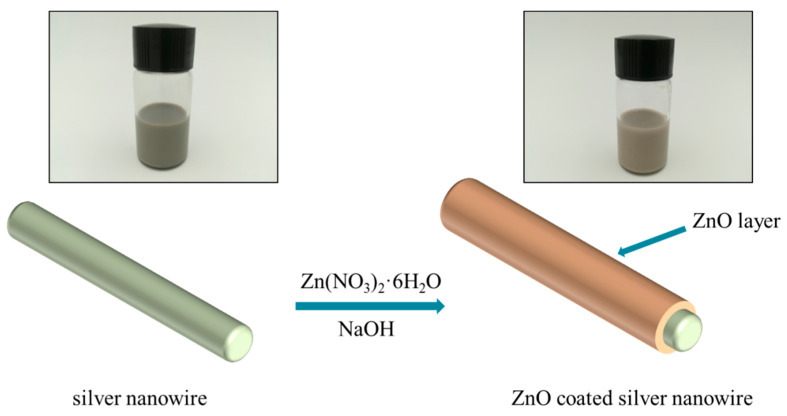
Schematic illustration of the preparation process of AgNW@ZnO filler.

**Figure 3 polymers-14-03539-f003:**
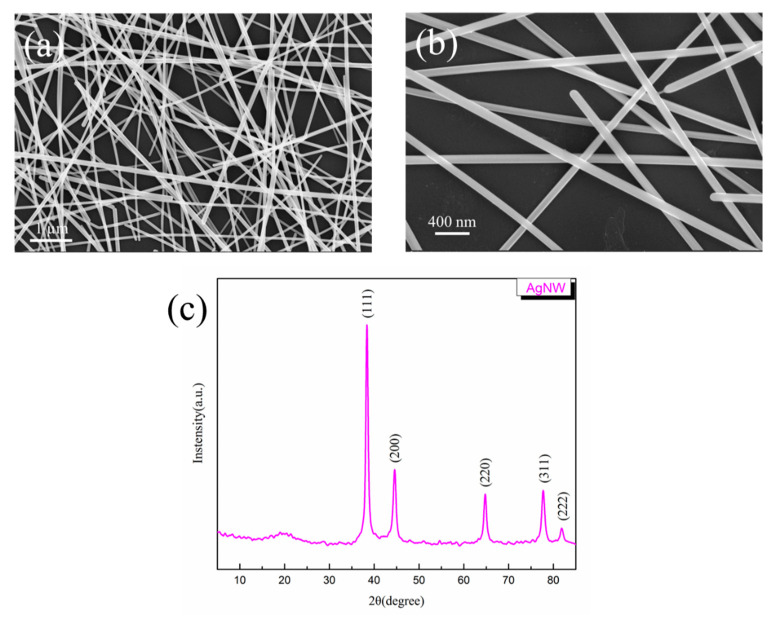
(**a**,**b**) are SEM images of the prepared AgNW; (**c**) XRD patterns of AgNW.

**Figure 4 polymers-14-03539-f004:**
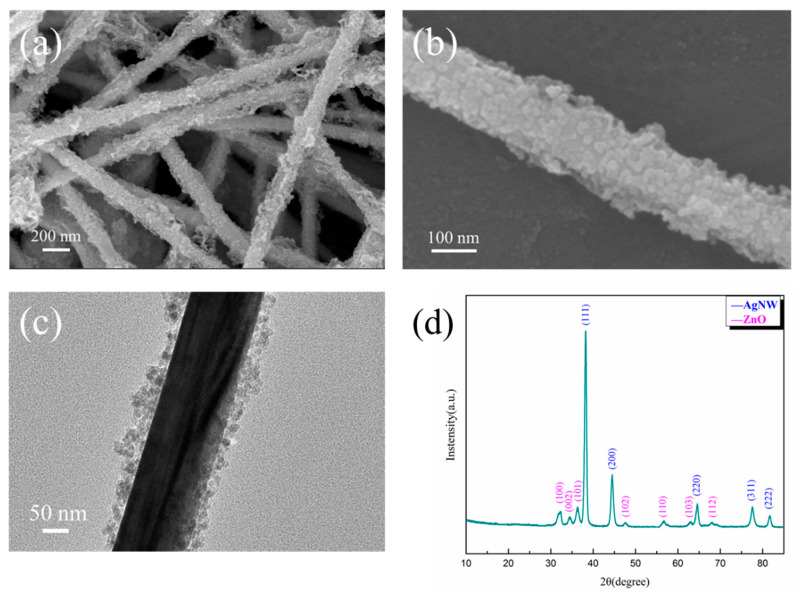
(**a**,**b**) are SEM images of the prepared AgNW@ZnO; (**c**) TEM image of AgNW@ZnO; (**d**) XRD pattern of AgNW@ZnO.

**Figure 5 polymers-14-03539-f005:**
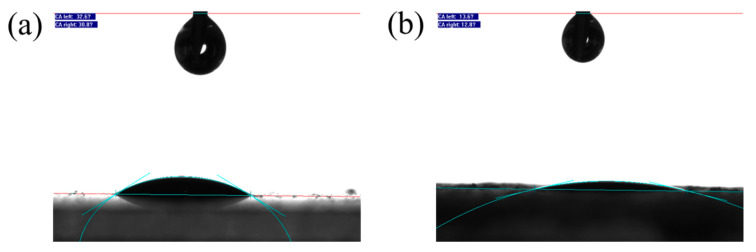
(**a**,**b**) are static contact angle test images of AgNW and AgNW@ZnO with epoxy.

**Figure 6 polymers-14-03539-f006:**
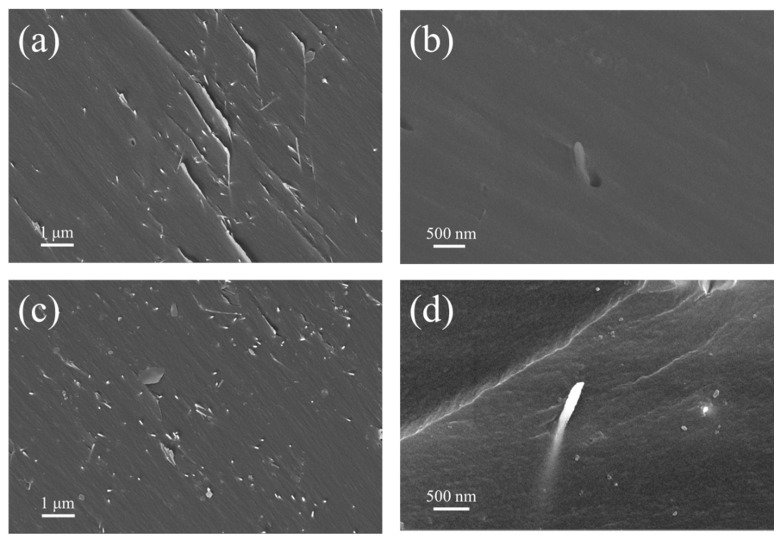
(**a**,**b**) are SEM photos of epoxy resin composite filled with AgNW; (**c**,**d**) are SEM photos of epoxy resin composite filled with AgNW@ZnO.

**Figure 7 polymers-14-03539-f007:**
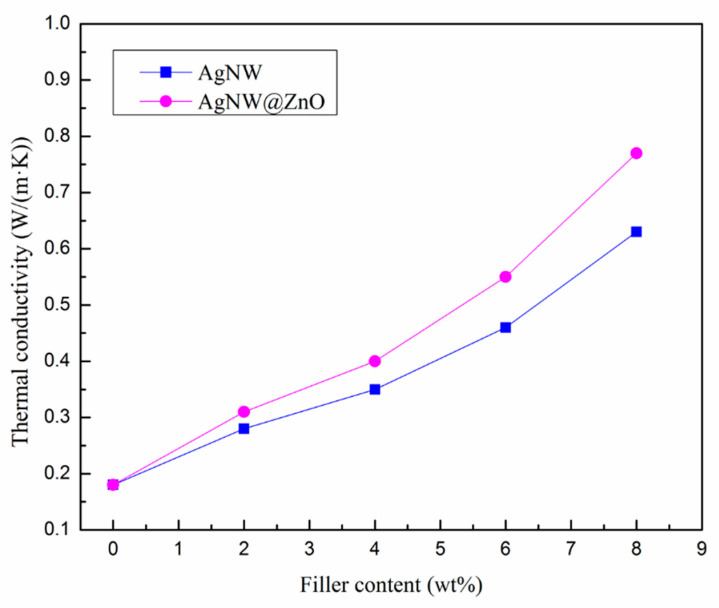
Thermal conductivity of AgNW and AgNW@ZnO-filled epoxy composites.

**Figure 8 polymers-14-03539-f008:**
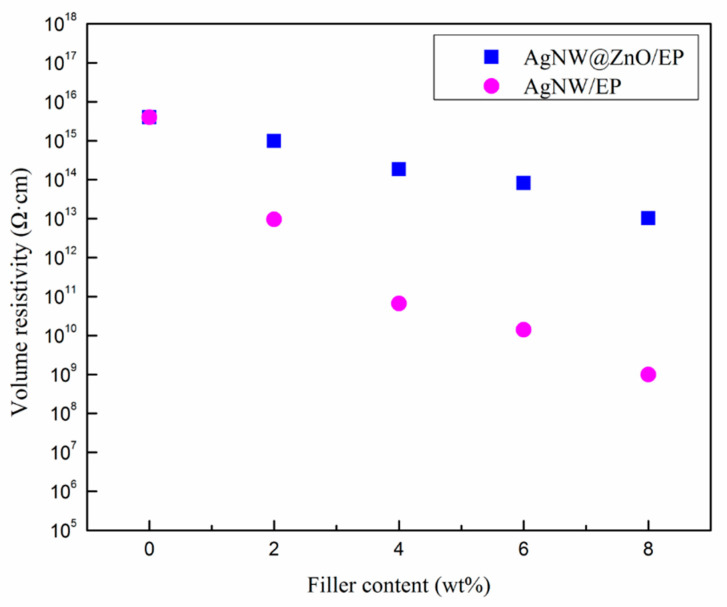
Volume resistivities of AgNW@ZnO/EP and AgNW/EP composites.

**Figure 9 polymers-14-03539-f009:**
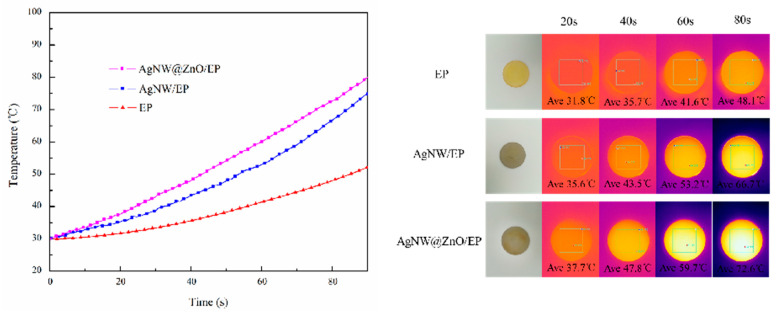
Surface temperature curves and corresponding infrared images of AgNW/EP and AgNW@ZnO/EP composites.

**Figure 10 polymers-14-03539-f010:**
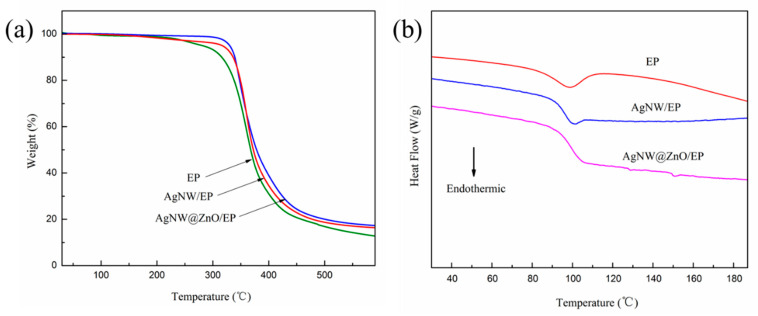
(**a**) TG and (**b**) DSC curves of EP, AgNW/EP, and AgNW@ZnO/EP composites.

**Figure 11 polymers-14-03539-f011:**
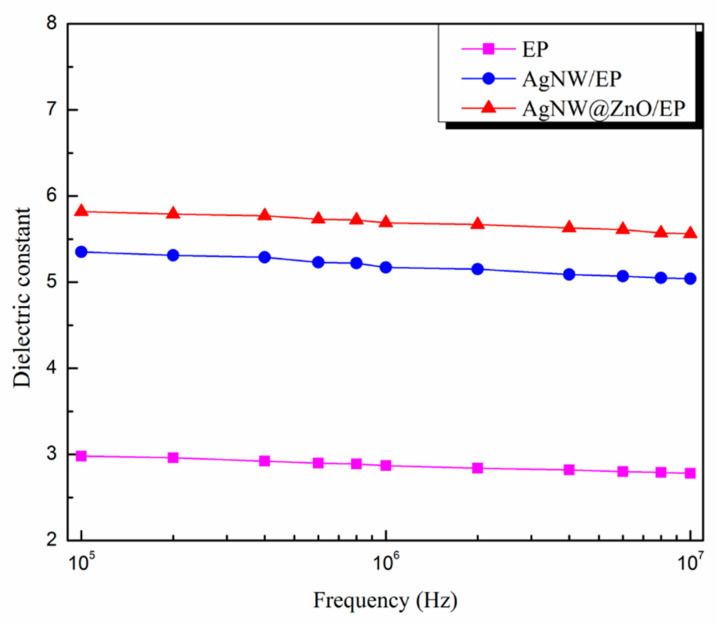
Dielectric constant of EP, AgNW/EP, and AgNW@ZnO/EP composites.

**Table 1 polymers-14-03539-t001:** Static contact angles of AgNW and AgNW@ZnO with epoxy.

Filler	AgNW	AgNW@ZnO
Static contact angle	31.7°	13.2°

**Table 2 polymers-14-03539-t002:** TG and DSC data of EP, AgNW/EP, and AgNW@ZnO/EP composites.

Sample	*T*_5%_ (°C)	*T*_50%_ (°C)	*T*_g_ (°C)
EP	271.8	372.1	91.3
AgNW/EP	300.2	374.9	96.1
AgNW@ZnO/EP	329.4	382.3	99.6

## Data Availability

The data presented in this study are available on request from the corrsponding author.

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
