# Peer review of "Highly Thermal Conductive and Electrically Insulating Epoxy Composites Based on Zinc-Oxide-Coated Silver Nanowires"

_polymers, 2022, doi:10.3390/polym14173539_

Round 1
Reviewer 1 Report
In this manuscript, the authors prepared AgNW@ZnO/EP conductive composites and tested their thermal conductivity and electrical insulation. The manuscript is overall well-written, I have some minor suggestions.
1. More references can be added.
(1) For example, in section 2.3, the authors mentioned that "There exist many methods to synthesize nano-ZnO in the literature, which include solid-phase reaction, hydrothermal process, precipitation technique, chemical vapor deposition, electrolysis, and magnetron sputtering." I suggest the authors to add a reference for each method mentioned.
(2) In addition, I suggest the authors to also add some references regarding the commonly used methods for the preparation of composite materials. For example, solution based self-assembly "Hybrid conjugated polymer/magnetic nanoparticle composite nanofibers through cooperative non-covalent interactions." Nanoscale Advances 2.6 (2020): 2462-2470.; Sol-gel method: "Improving piezoelectric properties of PVDF fibers by compositing with BaTiO3-Ag particles prepared by sol-gel method and photochemical reaction." Journal of Alloys and Compounds 883 (2021): 160810.
2. Please specify the crystal structure of the synthesized Ag NW.
3. The size of ZnO nanoparticle in the current study is about 5-25 nm, can the authors comment on how the size of the nanoparticle would affect the properties of the composites?
4. The Figure 6 captain has a typo: Figure 6 (a) and (b) are SEM photos of epoxy resin composite filled with AgNW; (b) and (c) are SEM photos of epoxy resin composite filled with AgNW@ZnO; Should (c) and (d) be the photos of epoxy resin composite filled with AgNW@ZnO instead of (b) and (c)?
Reviewer 2 Report
This paper describes highly thermal conductive and electrically insulating epoxy composites using nanofillers with high thermal conductivity. The authors deposited nano-ZnO particles on the surface of silver nanowires (AgNWs) by precipitation method and they added the obtained AgNW@ZnO filler to epoxy resin to improve the thermal conductivity of the composites. The authors proposed a new filler with core-shell structure. Especially, they found that combination of AgNW with ZnO (less thermal conductive than AgNW) resulted in increase of thermal conductivity. I think the experiments were carefully done and the results are reliable. The results of this paper will give useful information in the field of thermal interface materials for electronic components. I would like to accept this manuscript in Polymers.
May I have some comments.
- Figure 7 shows that thermal conductivity increases with filler content. I am just curious what will happen if the authors add more filler above 9 wt%.
- There is Chinese term in Table 2 (product ?)
- Caption of Table 2 should be revised (GO?).
